# Multitarget Action of Xanthones from *Garcinia mangostana* against α-Amylase, α-Glucosidase and Pancreatic Lipase

**DOI:** 10.3390/molecules27103283

**Published:** 2022-05-20

**Authors:** Juan Cardozo-Muñoz, Luis E. Cuca-Suárez, Juliet A. Prieto-Rodríguez, Fabian Lopez-Vallejo, Oscar J. Patiño-Ladino

**Affiliations:** 1Departamento de Química, Facultad de Ciencias, Universidad Nacional de Colombia, Sede Bogotá, Bogotá 111321, Colombia; jcardozom@unal.edu.co (J.C.-M.); lecucas@unal.edu.co (L.E.C.-S.); fhlopezv@unal.edu.co (F.L.-V.); 2Departamento de Química, Facultad de Ciencias, Pontificia Universidad Javeriana, Bogotá 110231, Colombia; juliet.prieto@javeriana.edu.co

**Keywords:** *Garcinia mangostana*, α-amylase, α-glucosidase, pancreatic lipase, digestive enzymes, xanthones, type 2 diabetes, obesity

## Abstract

Digestive enzymes such α-amylase (AA), α-glucosidase (AG) and pancreatic lipase (PL), play an important role in the metabolism of carbohydrates and lipids, being attractive therapeutic targets for the treatment of type 2 diabetes and obesity. *Garcinia mangostana* is an interesting species because there have been identified xanthones with the potential to inhibit these enzymes. In this study, the multitarget inhibitory potential of xanthones from *G. mangostana* against AA, AG and PL was assessed. The methodology included the isolation and identification of bioactive xanthones, the synthesis of some derivatives and a molecular docking study. The chemical study allowed the isolation of five xanthones (**1**–**5**). Six derivatives (**6**–**11**) were synthesized from the major compound, highlighting the proposal of a new solvent-free methodology with microwave irradiation for obtaining aromatic compounds with tetrahydropyran cycle. Compounds with multitarget activity correspond to **2**, **4**, **5**, **6** and **9**, highlighting **6** with IC_50_ values of 33.3 µM on AA, 69.2 µM on AG and 164.4 µM on PL. Enzymatic kinetics and molecular docking studies showed that the bioactive xanthones are mainly competitive inhibitors on AA, mixed inhibitors on AG and non-competitive inhibitors on PL. The molecular coupling study established that the presence of methoxy, hydroxyl and carbonyl groups are important in the activity and interaction of polyfunctional xanthones, highlighting their importance depending on the mode of inhibition.

## 1. Introduction

Obesity and diabetes are recognized as one of the main health problems worldwide, given their close relationship with other pathological disorders [1,2]. Diabetes mellitus is a metabolic disorder characterized by chronic hyperglycemia caused by a deficiency of the hormone insulin or by ineffective use of insulin in the body [1,2,3,4]. According to the World Health Organization (WHO), the prevalence of diabetes increased approximately four times, from 108 million diabetics in 1980 to 422 million in 2014. In 2019, this disease was the ninth leading cause of death, causing 1.5 million deaths [5]. By 2040, it is estimated that there will be 642 million people that suffer from diabetes [6]. More than 95% of people with diabetes have type 2 diabetes (T2D), a disease that results from the body’s ineffective use of insulin and is largely the result of excess body weight and physical inactivity [6,7,8]. A promising approach for the treatment of T2D is to decrease postprandial hyperglycemia by inhibiting carbohydrate hydrolyzing enzymes in the gastrointestinal tract. In this sense, the inhibition of enzymes such as α-amylase and α-glucosidase would lead to slower digestion of carbohydrates and a reduction in glucose absorption, attenuating postprandial glucose levels [9,10].

A pathology closely related to T2D is obesity, reporting that around 80% of diabetics (T2D) have or have been obese [2,6,11,12]. In 2016, a study documented that from 1974 to 2014 the prevalence of obesity increased sixfold, reaching 640 million adults with this condition [13]. According to the WHO, in 2016 more than 1.9 billion adults were overweight, and more than 650 million were obese [14]. Obesity is a non-communicable degenerative chronic disease in which the nutritional imbalance generates excess accumulated fat in the body, developing hypertrophy of adipose tissue [11,12,13,14]. Changes in eating habits, increased physical activity and taking supplements and medications are among the main approaches to treating this disease. Antiobesity medications generally involve three mechanisms: enhancement of energy expenditure, appetite suppression and inhibition of gastrointestinal enzymes involved in digestion, absorption and metabolism of carbohydrates and fats [9,10,15,16]. Since pancreatic lipase plays an important role in fat absorption, its inhibition has also been of great interest in drug development [15,16,17].

Some medicines for the treatment of obesity and diabetes have a mechanism of action based on the inhibition of digestive enzymes (orlistat which inhibits PL, acarbose and miglitol which inhibit AA and AG). However, many of these products have been associated with some gastrointestinal problems including bloating, diarrhea, flatulence, and abdominal discomfort [8,11,15]. Additionally, many of these drugs were designed with the aim of selectively inhibiting a single therapeutic target. Currently, many drug design studies are focused on finding compounds capable of exerting numerous physiological actions, especially for diseases of complex etiology [18,19,20]. Therefore, it is necessary to develop new efficient and safe multitarget agents to inhibit these types of enzymes that participate in the metabolism of carbohydrates and fats. Natural products can be an excellent alternative for the search for multitarget ligands since they produce a wide variety of chemical substances, many of which have been shown to have inhibitory activity on digestive enzymes [21,22,23].

*Garcinia mangostana* L (Clusiaseae), known as mangosteen, is a tropical tree native to Southeast Asia, which is cultivated in many tropical regions of the planet due to its economic importance [24]. The fruit is known as “the queen of fruits” for being juicy, soft and sweet, and its seeds and hulls have been used in traditional medicine to treat gastrointestinal disorders, cystitis, skin disorders, wound healing, ulcers, and weight loss [25,26,27,28]. In recent years, multiple in vivo and in vitro studies have been reported on mangosteen, highlighting its antioxidant, antibacterial, antifungal, antimalarial, anticancer, antinociceptive, anti-inflammatory, neuroprotective, antiobesity and antihyperglycemic activities [29,30].

Xanthones are the most common compound in *G. mangostana*, with around 90 xanthones reported, with a variety of substituents, such as hydroxyl, methoxyl, prenyl and glycosyl. Some are in the form of complex units, such as dimers, polycyclic or dibenzo-𝑦-pyrone derivatives [26,28]. These compounds affect multiple signaling pathways involved in different pathologies and turn them into a valuable source for developing new drugs to treat chronic diseases, such as obesity and type 2 diabetes. Different studies have reported the inhibitory capacity of some xanthones against enzymes such as PL, AA, and AG. For instance, α-mangostin and γ-mangostin inhibit PL with IC_50_ of 5.0 and 10.0 µM, respectively. Garcimangophenones A and B showed moderate inhibitory activity against α-amylase with IC_50_ of 9.3 and 12.2 µM. Furthermore, mangoxanthone A showed inhibitory activity against α-glucosidase and α-amylase with IC_50_ of 29.06 and 22.74 μM, respectively [31,32,33,34]. Even though these studies have demonstrated the potential of xanthones to inhibit PL, AG and AA separately, little has been done about their multitarget potential [25,26,30]. This work contributes to the search for molecules with multitarget inhibitory potential against pancreatic lipase, α-amylase and α-glucosidase, based on a study of the isolated xanthones from *G. mangostana* fruits and some derivatives such as AA, AG and PL inhibitors.

## 2. Results and Discussion

### 2.1. Phytochemical Study

A screening study was carried out on the mangosteen fruit pericarp. The ethanolic extract (EE) and VLC fractions (DCM, EtOAc, IPA, EtOH:H_2_O) were subjected to the enzyme inhibition test against PL, AA and AG (Table 1). The results indicated that EE and the DCM fraction are the most promising to inhibit the PL, AA and AG enzymes. This is the first report of inhibitory activity of EE on AG, while its activity had already been reported on PL (IC_50_ 26.50 ± 1.15 µg/mL) and AA (IC_50_ 105.36 ± 2.73 µg/mL) [17]. The reported results are different from those found in this investigation, observing that EE is more active on PL and less active on AA. The variations found in the results may be related to differences in the origin of the sample and experimental variations in the bioassays, which include variation in the reaction times, the substrates and the reading wavelengths [26,35]. The DCM fraction was selected to continue with the process of isolation and identification of the bioactive constituents, based on the fact of its potential multitarget activity.

The phytochemical study carried out on the DCM fraction from the pericarp of the *G. mangostana* fruit led to the isolation of five xanthones known as 9-hydroxycalabaxanthone **1**, 8-deoxygartanin **2**, gartanin **3**, α-mangostin **4** and γ-mangostin **5** (Figure 1). These compounds have been previously isolated from different organs of *G. mangostana* [36,37,38]. 9-hydroxycalabaxanthone **1** has been found in the Clusiaceae and Hypericaceae families and has been characterized by its anticancer, antioxidant, and ability to inhibit the PL and AG enzymes [34,39,40]. 8-deoxygartanin **2** has been reported in other species of the Clusiaceae family and has been found to have anticancer, anti-Alzheimer’s properties, and the ability to inhibit the PL and AG enzymes [34,39,40]. Gartanin **3** has been reported to be found in other species of the Clusiaceae family and multiple studies have shown its anticancer and antitumor properties, and ability to inhibit enzymes such as PL and AA [34,41,42]. α-mangostin **4** and γ-mangostin **5** have been isolated from species of the Clusiaceae and Hypericaceae families, and are characterized by their anti-Alzheimer’s, anticancer, anti-inflammatory, anti-diabetic and antiobesity properties [26,41,43,44,45,46,47,48].

### 2.2. Synthesis of Derivates

From the major constituent **4,** the synthesis of some derivatives was carried out to establish some preliminary structure–activity relationships (Figure 1). All the synthesized compounds (**6** to **11**) have been previously reported in the literature in species such as *G. fusca*, *G. mangostana* and *Cratoxylum cochinchinense*, and these were identified as fuscaxanthone C **6** [46], 3-isomangostin **7** [37], BR-xanthone A **8** [49], tetrahydro-α-mangostin **9** [50], 3,6-di-pentoxy-α-mangostin **10** [51] and 3,6-di-methoxy-4-methyl-α-mangostin **11** [52]. In the literature, it is reported that these xanthones have anticancer and antibacterial properties [37,46,51,52].

In this study, the synthesis of **7**, **8** and **11** with the methodologies used is reported for the first time. Compound **4** was subjected to a tetrahydropyran cycle formation reaction using microwave irradiation in the presence of pyridine hydrochloride. The reaction led to the formation of **7** and **8** which are characterized by the presence of tetrahydropyran type rings, which have been previously obtained as secondary products in the synthesis of γ-mangostin [49]. The methodology developed in the present investigation represents an interesting alternative to obtaining aromatic compounds with a tetrahydropyran cycle from hydroxylated aromatic precursors with a prenylated chain under solvent-free reaction conditions and microwave irradiation. On the other hand, when **4** was subjected to conditions for a typical methylation reaction in the presence of methyl iodide, potassium carbonate and silver oxide, the formation of the desired product **6** was observed, along with a product in low yield denominated **11**. This product is characterized by the C-methylation occurring, which could be favored by the presence of oxygenated activating groups present in the aromatic ring. The methylation reaction was carried out more efficiently and with higher yields to obtain **6** when carrying out the reaction without the presence of silver oxide (route i, Figure 1).

### 2.3. Determination of Enzymatic Inhibition against PL, AA and AG

The isolated and synthesized compounds (**1**–**11**) were evaluated as inhibitors of PL, AA and AG enzymes. For the compounds that presented potential inhibitory activity, the mean inhibitory concentration (IC_50_), the inhibition constants and the type of inhibition exerted on each enzyme were estimated (Table 2). Regarding PL, it was found that of the 11 compounds evaluated, 8 were able to inhibit the enzyme with IC_50_ values between 50.6 and 454.6 μM. However, neither compound achieved inhibition comparable to that caused by the positive control Orlistat. Compounds **7**, **8** and **10** were the substances that did not cause any inhibition in the catalytic activity of the enzyme. This study constitutes the first report of inhibitory activity against PL for compounds **6**, **9** and **11**. In previous studies, activity against PL has been reported for **1** (IC_50_ = 12 µM), **2** (IC_50_ = 50 µM), **3** (IC_50_ = 12 µM), **4** (IC_50_ = 5 µM; IC_50_ = 47.3 µM), and **5** (IC_50_ = 10 µM) [17,37]. Results reported by Chae and collaborators and those obtained in this study are not comparable, because very different methodologies were used [34]. On the other hand, the results obtained for **4** agree with those reported by Adnyana and collaborators, since the same method of quantifying the enzymatic activity was used [17]. A preliminary structure–activity analysis indicates that compounds having isoprenyl chains at positions 2 and 8 of the xanthone nucleus (**4**, **5**, **6** and **11**) have the lowest IC_50_ values, suggesting that the presence and location of these substituents are important for PL inhibition. It is important to highlight that even though compound **10** has isoprenyl substituents in positions 2 and 8, it does not show activity on PL, indicating that the O-alkylation on the hydroxyls in positions 3 and 6 negatively influences the activity. Another aspect that can be highlighted is the negative effect that the hydrogenation of the prenylated chains has on the inhibitory activity on PL, since when comparing the IC_50_ value of **4** with that of **9**, the activity is reduced by approximately one order of magnitude. The kinetic study determined that the compounds **3** and **5** are uncompetitive inhibitors, **1**, **2** and **4** are non-competitive inhibitors, while **6**, **9** and **11** are mixed inhibitors against PL, in accordance with previous reports in the literature [40]. In this case, it was found that all the estimated inhibition constants are greater than Km (47.4 µM), therefore, it is possible to affirm that the inhibitors have less affinity with the enzyme than the substrate used in the assay. The types of inhibition caused by **1**, **2**, **3**, **5**, **6**, **9** and **11** on PL are reported for the first time.

The results of the α-amylase inhibition study reveal that compounds **2**, **4**, **5**, **6** and **9** caused AA inhibition with IC_50_ values between 90.1 and 333.5 μM, showing much higher activity than that caused by the positive control acarbose. In previous studies, the activity against AA of compounds **4** (IC_50_ of 29.6 µM [17]; 60% inhibition at a concentration of 100 µM [42]) and **5** (67.3% inhibition at a concentration of 100 µM [42]) was determined. The differences found are related to the methodologies used, since both our study and that of the other two authors use enzymes of different origin, substrates, reaction times and different quantification methods [17,35,42]. This is the first report of inhibitory activity against AA for compounds **2**, **6** and **9**. A preliminary study of relations of structure-inhibitory activity on AA showed that the methylation of the phenolic hydroxyls favors the activity, which is evidenced by comparing the IC_50_ values of **6**, **4** and **5**, being **6** approximately two times more active than **4** and **5**. It was also observed that the hydrogenation of the prenylated chains increases the inhibitory activity on AA, since **9** is approximately three times more active than **4**. Comparing the results of inhibitory activity of **11** and **6**, it is found that methylation at position 4 of the xanthone nucleus completely annuls the activity on the enzyme. Finally, it can be established that the pyran-type rings present in the structures of **1**, **7** and **8** cause the inhibitory activity on AA to be completely lost. The kinetic study made it possible to determine that compound **2** is a non-competitive type of inhibitor, while **4**, **5**, **6** and **9** are competitive type inhibitors. The Ki values determined for all inhibitors are less than Km (340 µM), indicating that they have a higher affinity for the enzyme than the substrate used in the assay. It should be noted that the types of inhibition determined for bioactive xanthones on AA are reported for the first time in this study.

Regarding AG, it is observed that of the 11 compounds evaluated, 8 were able to inhibit the enzyme with IC_50_ values between 33.3 and 247.8 μM. All active compounds had a greater inhibitory effect on AG than the positive control acarbose, being the compounds **4**, **6** and **7** approximately 10 times more active than the positive control. Compounds **8**, **10** and **11** were the substances that did not cause any inhibition in the catalytic activity of AG. In previous studies, the activity for compounds **1** to **5** against AG was reported, finding that the results obtained in this study differ from previous reports. Li and collaborators report IC_50_ values of 10.0 µM for **4** and 5.0 µM for **5** [10]. Ryu and collaborators report IC_50_ values of 34.2 µM for **1**, 21.5 µM for **2**, 10 µM for **3**, 5 µM for **4**, and 1.5 µM for **5 [39]**. In the study developed by Vongsak and collaborators., they report IC_50_ values of 29.2 µM for **4** and 4.2 µM for **5 [53]**. In all studies the method used to quantify the enzymatic activity of AG was the same, however, the reaction times and substrate concentrations are different in all cases. This is the first report of inhibitory activity against AG for compounds **6**, **7** and **9**. Comparing the IC_50_ values of **4** and **5**, it is possible to establish that the presence of a methoxy group in position 7 of the xanthone nucleus favors the inhibitory activity on AG, being **4** approximately three times more active than **5**. When comparing the activity of **7** and **8**, it is observed that the presence of the pyran ring in positions 7 and 8 of the xanthone nucleus cancels the activity on AG. Another structural characteristic that leads to the loss of activity on this enzyme is the presence of a methyl group in position 4 of the xanthone, an effect that is confirmed when comparing the results obtained for **6** and **11**. Finally, comparing the activity exhibited by **7** and **1**, it is possible to appreciate that the activity is reduced approximately four times when the pyran ring located in positions 2 and 3 of the xanthone has an unsaturation that extends the conjugation of the aromatic ring. The kinetic study was able to determine that the compounds **2** to **5** and **7** are mixed-type inhibitors of AG, as reported in the literature [39]. Compound **1** is an uncompetitive inhibitor, while **6** and **9** are competitive inhibitors. All the estimated inhibition constants are greater than Km (12 µM), therefore, it is possible to affirm that the inhibitors have a lower affinity for the enzyme than the substrate used in the assay. The types of inhibition caused by **1**, **3**, **4**, **6**, **7** and **9** on AG are reported for the first time.

### 2.4. Molecular Docking Studies

The protein–ligand docking analysis, for PL, AA, AG, and all compounds (**1–11**) was carried out with AutoDock Vina [54] and the visualization with Free Maestro [55]. All binding energies of compounds **1**–**11** are summarized in Appendix A (Appendix A). The binding site selection (orthosteric or allosteric) was guided by the experimental type of inhibition (Table 2), and blind molecular docking, as shown for compound **4** (Figure 2).

The molecular docking studies allowed us to illustrate the key role that some groups have on the xanthone scaffolds in the interaction with PL, AA and AG, which could explain some structure–activity relationships. To summarize the most relevant findings from molecular docking studies, Figure 3 shows the 2D representation of the predicted binding modes for the top two active molecules for each target (compounds **4**, **5**, **6** and **9**). These binding modes correspond to the lowest binding energies (DG) found for the selected binding site (orthosteric or allosteric).

The results of the blind molecular docking study from PL show that xanthones have a preference for a binding site different from the catalytic pocket (oxyanion hole) that contains the amino acids Ser^152^(OH), Phe^77^(NH) and His^263^(NH) [56]. These results agree with enzyme kinetics studies since the compounds **1**–**11** showed a non-competitive and an uncompetitive inhibition. The molecular docking model (Figure 2) indicates that there is an existing pocket at the interface between PL and colipase where the molecules **1**–**11** exhibit higher binding affinities. The main interaction observed in the PL-colipase pocket is pi-pi type with the amino acid Tyr^369^ of colipase, which suggests that this pocket is involved in the decrease of catalytic activity. Compounds may be causing a conformational change in the catalytic site since this is formed as the PL-colipase heterodimer is formed [56]. In the case of **6** and **11**, whose inhibition was of a mixed type, the results of molecular docking suggest that their inhibition character tends towards the non-competitive type of inhibition, since no interaction with the catalytic triad of PL (Figure 3a,b).

The blind molecular docking study on AA shows that bioactive xanthones mainly dock on the catalytic site. For the compounds **4**, **5** and **9**, hydrogen bonds interactions are observed with the amino acid residues Asp^197^(OH), Glu^233^(OH), Arg^195^(NH_2_), interacting with at least one of the amino acid residues of the catalytic triad, confirming the competitive inhibition of these compounds. It is important to point out that compound **9** has a hydrogen bond interaction with ASP^300^(OH), an amino acid residue of the catalytic site responsible for substrate anchoring and subsequent sugars cleavage [57], which corroborates the competitive inhibition of this compound since it blocks the catalytic site and competes with the substrate. On the other hand, compound **6**, despite being in the catalytic site, does not show interaction with the amino acids of the catalytic triad or with amino acids close to it, which suggests that the mixed inhibition of this compound has a greater tendency to non-competitive inhibition. This may be related to the presence of methoxy groups in positions 4, 6 and 7 that modify their arrangement in space and therefore do not allow interactions to be established with the catalytic triad (Figure 3c,d). Lastly, compound **2** has interactions with the amino acids Thr^163^, Trp^58^ and Trp^59^ that do not belong to the catalytic triad, which confirms its non-competitive inhibition mode.

The AG molecular docking models show that compounds **6** and **9** dock in the catalytic pocket, but do not interact with the amino acids of the catalytic triad, Asp^203^(OH), Asp^327^(OH) and Asp^443^(OH) [58], suggesting that these compounds are exerting competitive inhibition by sterically blocking catalytic pocket. Furthermore, compound **4** was also docked with an amino acid close to the anchor amino acid in the catalytic site, so, sterically blocking the catalytic pocket as we observed in docking studies, indicating the possibility that it is a mixed inhibitor. Unlike the above, it is observed that compounds **1**, **2**, **3**, **5** and **7** do not dock in the catalytic site, but rather interact in an allosteric site, close to the catalytic site, with the amino acid residues Asp^18^(OH), Glu^242^(OH) and Asp^474^(OH). Interactions with the allosteric pocket were also found for compound **4**. Given the proximity between the catalytic site, responsible for the cleavage of sugars, and the allosteric site, it can be inferred that the interaction of xanthones with the allosteric pocket would cause conformational change, impediment steric or occupation of the surrounding space of the catalytic site in the enzyme that can lead to the reduction in catalytic activity [42,56,58], a fact that confirms the behavior of these compounds as mixed and uncompetitive inhibitors (Figure 3e,f). It is worth mentioning that there are no previous reports in the literature for compounds **6**–**11** on their AG activity. The relevance of docking studies in an investigation of this type can be underlined, which, in combination with biological activity data, is particularly used. A combination of approaches, such as these, offers a broader perspective to explain the effects that compounds with polypharmacological properties produce on each target and may suggest useful structural features in the search for molecules of this nature.

### 2.5. Compounds with Polypharmacological Action

Multitarget drugs, which can simultaneously interact with several biological targets, lead to new and more effective drugs for a variety of complex diseases, even with relatively weak activities [25,26,34,39,42,47,59]. In this sense, when comparing the activity results of xanthones on the three digestive enzymes (PL; AA and AG) (Table 2), it was found that compounds **1** and **3** inhibit the catalytic activity of PL and AG and that the compounds **2**, **4**, **5**, **6** and **9** exert an inhibitory effect on the three enzyme targets. Compounds **4** and **6** are those with the greatest multitarget potential since they present a moderate activity on PL and an activity on AG approximately ten times greater than positive control of acarbose. For the AA enzyme, it is observed that compound **6** is five times more active than the positive control, while compound **4** is approximately three times more active than acarbose. It is important to note that compound **9**, despite having low activity on the PL enzyme, exhibits a promising inhibition profile on the two glycoside hydrolases enzymes, since on AA and AG the activity of this compound is approximately ten times higher than the positive control acarbose. The foregoing suggests that xanthones could be compound leads proposed as candidates for a structural base to multidrug drugs for the development of targeted treatments in type 2 diabetes and obesity, where possible keeping to interaction with amino acids of each enzyme and the specific sites of a molecule with methoxy, hydroxyl and prenyl chains groups to which the decrease in catalytic activity is attributed. These results also demonstrate the polypharmacological potential of xanthones from *G. mangostana* to inhibit the PL, AG and AA enzymes in a multitarget manner.

## 3. Materials and Methods

### 3.1. General Experimental Procedures

All commercially available reagents employed were used without further purification, while the solvents were technical grade and distilled before use. Thin-layer chromatography (TLC) was performed on SiliaPlateTM alumina plates pre-coated with silica gel 60 F_254_ (SiliCycle^®^ Inc., Quebec City, QC, Canada). Vacuum Liquid Chromatography (VLC) was performed on SiliaPlateTM silica gel F_254_ of size 5–20 µm (SiliCycle^®^ Inc., QC, Canada). Flash Chromatography (FC) was performed on SiliaFlash^®^ silica gel P60 of size 40–63 µm (SiliCycle^®^ Inc., QC, Canada). Melting points were recorded on a Thermo Scientific 00590Q Fisher-Johns apparatus (Thermo Scientific^®^, Waltham, MA, USA). NMR measurements were performed on Bruker Advance AC-400 spectrometer (Bruker^®^, Hamburg, Germany) ^1^H-NMR and APT experiments, operating at 400 MHz for ^1^H and 100 MHz for APT; ^1^H-^1^H, direct ^1^H-^13^C and long-range ^1^H-^13^C scalar spin–spin connectivity was established by 2D spectroscopic analysis of the COSY, HMQC and HMBC experiments. Chemical shifts (δ) were reported in part per million (ppm) and coupling constants (J) in Hz. The following abbreviations were used to designate chemical shift multiplicities: s = singlet, d = doublet, t = triplet, q = quartet, m = multiplet, bs = broad singlet. 

The enzymes used for the enzyme inhibition studies were pancreatic lipase type II (PL) from porcine pancreas (100–400 units/mg protein, L-SLBD2433V, Sigma-Aldrich, EC. 3.1.1.3); α-amylase (AA) type VI-B from porcine pancreas (≥10 units/mg solid, L-SLBP4061V, Sigma-Aldrich, EC. 3.2.1.1) and α-glucosidase (AG) type I from Saccharomyces cerevisiae (lyophilized powder, ≥10 units/mg protein, L-SLBX6245, Sigma-Aldrich, EC 3.2.1.20). The following compounds were used as substrates: 4-nitrophenyl dodecanoate (Sigma-Aldrich) for PL, soluble potato starch (Sigma-Aldrich) and 3,5-dinitrosalicylic acid (DNS, as derivatizing reagent) were used for AA, and 4-nitrophenyl-α-D-glucopyranoside (Sigma-Aldrich) for AG. Absorbance readings were performed on a Thermo Scientific Multiskan GO microplate reader (Waltham, MA, USA) using Skanlt RE4.1 software.

### 3.2. Plant Material

Mangosteen fruit was acquired in the Paloquemao marketplace, Bogotá (Colombia). The taxonomic determination of the species was carried out by the biologist Néstor García in the herbarium of the Pontificia Universidad Javeriana with the collection number HPUJ-29768.

### 3.3. Isolation and Identification of Xanthones from G. mangostana

The dried and ground pericarp of the fruit of *G. mangostana* (3345 g) was extracted with ethanol at 96% at room temperature by the maceration method. The solvent was evaporated by reduced pressure distillation and a dry ethanolic extract (EE, 572.2 g) was obtained. A part of the EE (565 g) was fractionated by vacuum liquid chromatography (VLC) using solvents of different polarity: dichloromethane (212.5 g), ethyl acetate (210.0 g), isopropanol (108 g) and ethanol:water 80:20 (31.6 g). The resulting fractions were tested as inhibitors of catalytic activity against PL, AA and AG, establishing the fraction of dichloromethane (DCM) as the one with the highest inhibitory activity. The DCM fraction (65.1 g) was subjected to flash chromatography (FC) using as mobile phase in gradient hexane:EtOAc 95: 5 to hexane:EtOAc 50:50, resulting 4 fractions (D1, D2, D3 and D4) according to the profile presented in TLC. Fraction D1 (40.3 g) was purified by FC eluting with hexane:EtOAc in increasing polarity 95:5 at 70:30, obtaining seven fractions (D1.1 to D1.7). Fraction D1.2 (437 mg) was purified by FC eluting with hexane:EtOAc (95:5) obtaining a yellow solid denominated 9-hydroxylabaxanthone (**1**, 191 mg, m.p. 155–157 °C). Fraction D1.3 (3.3 g) was subjected to FC eluting with hexane:EtOAc (90:10 to 80:20) obtaining a green solid identified as 8-deoxygartanin (**2**, 125 mg, m.p. 165–167 °C) and a yellow solid identified as gartanin (**3**, 65 mg, mp: 166–168 °C). Fraction D1.4 (18.6 g) was purified by FC with hexane:EtOAc (80:20 to 60:40), obtaining a yellow solid denominated α-mangostin (**4**, 3.0 g, m.p. 180–182 °C). Fraction D2 (14.8 g) was subjected to FC purification with elution system hexane:EtOAc (80:20 to 70:30), obtaining seven fractions (D2.1 to D2.7). Fractions D2.1 and 2.2 (764 mg) were purified by successive FC eluting with hexane:EtOAc (90:10 to 80:20) obtaining compound **3** (112 mg). Fractions D2.3 to D2.5 were pooled (7.5 g) and subjected to FC with the elution system hexane:EtOAc (80:20) and subsequently with hexane:EtOAc (60:40), obtaining the compound **4** (1.0 g). Fraction D3 (6.9 g) was purified by successive FC with hexane:EtOAc (70:30 to 50:50), obtaining compound **4** (329 mg) and a brown crystalline solid identified as γ-mangostin (**5**, 131 mg, m.p. 206–208 °C).

9-hydroxicalabaxantone (**1**): Yellow solid, melting point (m.p.): 155–157 °C. ^1^H-NMR (CDCl_3_, 400 MHz): ẟ (ppm) 13.69 (s, 1H), 6.83 (s, 1H), 6.73 (d, *J* = 10.1 Hz, 1H), 6.37 (s, 1H), 6.24 (s, 1H), 5.56 (d, *J* = 10.1 Hz, 1H), 5.26 (s, 1H), 4.09 (d, *J* = 6.2 Hz, 2H), 3.80 (s, 3H), 1.83 (s, 3H), 1.69 (s, 3H), 1.46 (s, 6H). APT (100 MHz, CDCl_3_): ẟ (ppm) 182.0 (C-9), 159.9 (C-1), 157.9 (C-3), 156.8 (C-4′), 155.8 (C-5′), 154.5 (C-6), 142.5 (C-7), 136.9 (C-8), 132.1 (C-16), 127.1 (C-11), 123.1 (C-15), 115.7 (C-10), 111.9 (C-8´), 104.2 (C-2), 103.2 (C-9′), 101.6 (C-5), 94.1 (C-4), 77.9 (C-12), 62.0 (C-19), 28.3 (C-13, C-13´), 26.5 (C-14), 25.8 (C-17), 18.2 (C-18). The spectroscopic data were consistent with those reported in the literature for 9-hydroxicalabaxantone [36].

8-deoxygartanin (**2**): Green solid, m.p. 165–167 °C. ^1^H-NMR (CDCl_3_, 400 MHz): ẟ (ppm) 13.19 (s, 1H), 7.77 (dd, *J* = 7.9, 1.5 Hz, 1H), 7.30 (dd, *J* = 7.8, 1.6 Hz, 1H), 7.25 (s, 1H), 6.53 (s, 1H), 5,68 (br s, 1H), 5.32–5.23 (m, 2H), 3.56 (d, *J* = 7.0 Hz, 2H), 3.49 (d, *J* = 7.2 Hz, 2H), 1.88 (s, 3H), 1.86 (s, 3H), 1.79 (s, 3H), 1.76 (s, 3H). APT (100 MHz, CDCl_3_): ẟ (ppm) 180.9 (C-9), 160.7 (C-1), 158.5 (C-3), 152.2 (C-4′), 144.2 (C-5′), 144.1 (C-5), 136.0 (C-12′), 133.3 (C-12), 123.6 (C-7), 122.0 (C-11), 121.0 (C-11′), 120.7 (C-4), 119.5 (C-8), 116.7 (C-6), 108.9 (C-9′), 105.2 (C-2), 103.1 (C-9′), 25.7 (C-14), 25.4 (C-13), 21.9 (C-10′), 21.5 (C-10′), 17.7 (C-13′). The spectroscopic data were consistent with those reported in the literature for 8-deoxygartanin [60]. 

Gartanin (**3**): Yellow solid, m.p. 166–168 °C. ^1^H-NMR (CDCl_3_, 400 MHz): ẟ (ppm) 12.33 (s, 1H), 11.25 (s, 1H), 7.22 (d, *J* = 8.9 Hz, 1H), 6.66 (d, *J* = 8.9 Hz, 1H), 6.60 (s, 1H), 5.30–5.21 (m, 2H), 5.09 (s, 1H), 3.50 (d, *J* = 7.2 Hz, 2H), 3.46 (d, *J* = 7.2 Hz, 2H), 1.86 (s, 6H), 1.79 (s, 3H), 1.75 (s, 3H). APT (100 MHz, CDCl_3_): ẟ (ppm) 184.8 (C-9), 161.7 (C-1), 158.2 (C-8), 153.9 (C-3), 152.6 (C-5), 142.9 (C-5′), 136.4(C-12′), 135.8 (C-4′), 134.0 (C-12), 123.0 (C-7), 121.9 (C-11), 121.1 (C-11′), 109.9 (C-6), 109.6 (C-9′), 105.9 (C-8′), 102.3 (C-9′), 26.0 (C-13, C-14), 25.8 (C-14′), 22.1 (C-10), 21.7 (C-10′), 18.1 (C-13′). The spectroscopic data were consistent with those reported in the literature for gartanin [61].

α-mangostin (**4**): Yellow solid, m.p. 180–182 °C. ^1^H-NMR (CDCl_3_, 400 MHz): ẟ (ppm) 13.78 (1H, s), 6.83 (s, 1H), 6.30 (s, 2H), 6.14 (s, 1H), 5.37–5.21 (m, *J* = 13.5, 5.4 Hz, 2H), 4.09 (d, *J* = 6.0 Hz, 1H), 3.81 (s, 3H), 3.46 (d, *J* = 7.2 Hz, 1H), 1.84 (d, *J* = 4.8 Hz, 6H), 1.77 (s, 3H), 1.69 (s, 3H). APT (100 MHz, CDCl_3_): ẟ (ppm) 181.8 (C-9), 161.4 (C-1), 160.4 (C-3), 155.6 (C-6), 154.9 (C-5′), 154.3 (C-4′), 142.4 (C-7), 136.8 (C-12′), 135.7 (C-12), 131.9 (8), 122.9 (C-11), 121.2 (C-11′), 112.0 (C-9′), 108.2 (C-2), 103.5 (C-8′), 101.3 (C-5), 93.1 (C-4), 61.9 (C-15), 26.4 (C-13′, C-14′), 25.6 (C-14), 21.2 (C-10′), 18.0 (C-10), 17.8 (C-13). The spectroscopic data were consistent with those reported in the literature for α-mangostin [62]. 

γ-mangostin (**5**): Brown crystalline solid, m.p. 206–208 °C. ^1^H-NMR (CD_3_OD, 400 MHz): ẟ (ppm) 6.69 (s, 1H), 6.25 (s, 1H), 5.28 (s, 2H), 4.14–4.04 (d, *J* = 6.9 Hz, 4H), 3.31 (s, 2H), 1.86 (s, 3H), 1.80 (s, 3H), 1.68 (s, 6H). APT (100 MHz, CD_3_OD): ẟ (ppm) 182.1 (C-9), 161.9 (C-1), 160.1 (C-3), 154.8 (C-7), 152.6 (C-6), 151.8 (C-4′), 140.6 (C-5′), 130.3 (C-8), 130.2 (C-2), 128.1 (C-12 y C-12′), 123.4 (C-11′), 122.5 (C-11), 109.7 (C-9′), 102.4 (C-8′), 99.5 (C-5), 91.5 (C-4), 25.2 (C-13′), 24.6 (C-14′), 24.5 (C-10′), 20.8 (C-10), 16.9 (C-14), 16.5 (C-13). The spectroscopic data were consistent with those reported in the literature for γ-mangostin [63]. 

### 3.4. Preparation of Derivatives

Fuscaxanthone C (**6**). The methylation reaction on **4** was carried out in the presence of methyl iodide and potassium carbonate [64]. Excess CH_3_I (3.4 mL, 5.46 mmol) was added to a mixture of **4** (100 mg, 0.255 mmol), K_2_CO_3_ (310 mg, 2.24 mmol) and anhydrous acetone (5 mL). The reaction mixture was stirred under reflux for approximately 60 h and continuously monitored by TLC until the disappearance of **4**. Subsequently, the excess CH_3_I was removed by distillation under reduced pressure and H_2_O (10 mL) was added to the resulting residue. Then, a liquid–liquid extraction was performed with AcOEt (4 × 5 mL) and the organic phases combined were washed with saturated NaCl solution (3 × 5 mL), and dried with anhydrous Na_2_SO_4_. The crude product resulting from the vacuum concentration was purified by FC eluting with hexane:AcOEt (90:10), to give **6** (89.3 mg, 47.3% yield). Compound **6**: yellow solid, mp: 118–120 °C. ^1^H-NMR (CDCl_3_, 400 MHz): ẟ (ppm) 13.71 (s, 1H), 6.72 (d, *J* = 11.1 Hz, 1H), 6.32 (s, 1H), 5.32–5.15 (m, 2H), 4.16 (d, *J* = 6.6 Hz, 2H), 3.95 (s, 3H), 3.90 (s, 3H), 3.79 (s, 3H), 3.35 (d, *J* = 7.1 Hz, 2H), 1.85 (d, *J* = 18.9 Hz, 6H), 1.68 (s, 6H). APT (100 MHz, CDCl_3_): ẟ (ppm) 182.0 (C-9), 163.4 (C-1), 159.8 (C-3), 158.0 (C-6), 155.4 (C-5′), 155.3 (C-4′), 144.0 (C-7), 137.3 (C-8), 131.8 (C-12′), 131.2 (C-12), 123.2 (C-11), 122.3 (C-11′), 112.1 (C-2), 111.5 (C-8′), 104.0 (C-9′), 98.2 (C-5), 88.6 (C-4), 60.9 (C-17), 56.0 (C-16), 55.8 (C-15), 26.1 (C-14), 25.9 (C-13), 25.9 (C-10′), 21.3 (C-10′), 18.2 (C-14′), 17.8 (C-13′). The spectroscopic data were consistent with those reported in the literature for fuscaxanthone C [46].

3-isomangostin (**7**) and BR-xanthone-A (**8**). Compound **4** was subjected to a demethylation reaction using hydrochloride pyridine (py-HCl) under solvent-free conditions and microwave irradiation [65]. A mix of **4** (100 mg, 0.252 mmol) and py-HCl (180 mg, 1.25 mmol) were placed in a flask of 25 mL, and this was sealed. The mixture was subjected to microwave irradiation in conventional equipment with 20% power (215 W) for periods of 2 min until completing a total of 13 cycles (26 min). The reaction was monitored by TLC until verifying the disappearance of **4**. Subsequently, cold water (20 mL) was added and extracted with DCM (3 × 15 mL). The combined organic phases were washed with H_2_O (3 × 5 mL) and dried over anhydrous Na_2_SO_4_. The crude product resulting from the removal of the solvent by vacuum concentration was purified by FC eluting with hexane:AcOEt (90:10), to give 7 (63.4 mg, 63.3% yield) and **8** (23 mg, 23.0% yield).

Compound **7**: yellow solid, mp: 165–167 °C. ^1^H-NMR (CDCl_3_, 400 MHz): ẟ (ppm) 13.71 (s, 1H), 6.83 (s, 1H), 6.27–6.23 (m, 2H), 5.29 (d, *J* = 10.5 Hz, 1H), 4.12 (d, *J* = 6.2 Hz, 1H), 3.80 (s, 3H), 2.73 (d, *J* = 6.8 Hz, 2H), 1.85 (d, *J* = 6.4 Hz, 5H), 1.68 (s, 3H), 1.37 (s, 6H). APT (100 MHz, CDCl_3_): ẟ (ppm) 182.0 (C-9), 160.7 (C-1), 160.5 (C-6), 154.7 (C-5′), 154.4 (C-4′), 145.8 (C-3), 142.5 (C-7), 136.9 (C-8), 132.1 (C-16), 123.2 (C-15), 109.9 (C-8′), 103.5 (C-2), 102.5 (C-9′), 101.6 (C-5), 94.0 (C-4), 76.0 (C-12), 62.0 (C-19), 31.9 (C-11), 26.7 (C-13, C-13′), 26.5 (C-14), 25.8 (C-18), 18.2 (C-17), 16.1 (C-10). The spectroscopic data were consistent with those reported in the literature for 3-isomangostin [37].

Compound **8**: White solid, mp: 160–162 °C. ^1^H-NMR (CDCl_3_, 400 MHz): ẟ (ppm) 13.74 (s, 1H), 6.79 (s, 1H), 6.38 (s, 1H), 6.24 (s, 1H), 3.50 (s, 2H), 2.71 (s, 2H), 1.88 (s, 2H), 1.83 (s, 2H), 1.58 (s, 1H), 1.38 (d, *J* = 6.7 Hz, 12H). APT (100 MHz, CDCl_3_): ẟ (ppm) 182.6 (C-9), 160.5 (C-1), 160.4 (C-6), 154.9 (C-4′), 153.2 (C-5′), 151.5 (C-3), C-4′), 137.7 (C-7), 121.3 (C-8), 111.2 (C-8′), 103.5 (C-9′), 103.0 (C-2), 100.5 (C-5), 94.0 (C-4), 75.9 (C-12′), 75.5 (C-12), 32.8 (C-11), 31.9 (C-11′), 26.7 (C-13 y C-14), 26.4 (C-13′ y C-14′), 22.3 (C-10′), 16.0 (C-10). The spectroscopic data were consistent with those reported in the literature for BR-xanthone-A [49].

Tetrahydro-α-mangostin (**9**). The hydrogenation reaction on **4** was carried out in the presence of molecular hydrogen, palladium supported on carbon (Pd/C) and anhydrous methanol [66]. Pd/C (13.4 mg, 0.126 mmol) was added to a solution of **4** (200 mg, 0.504 mmol) in anhydrous methanol (5 mL), and its atmosphere was saturated with H_2_. The reaction mixture was stirred at room temperature for 60 h and continuously monitored by TLC until the disappearance of **4**. The resulting mixture was filtered on quantitative Whatman paper and the residue was washed with DCM (2 × 20 mL). The combined organic layer was dried over anhydrous Na_2_SO_4_ and concentrated under a vacuum. The crude product was purified by FC eluting with hexane:AcOEt (80:20), obtaining **9** (160.4 mg, 98.7% yield). Compound **9**: yellow solid, m.p.: 85–87 °C. ^1^H-NMR (CDCl_3_, 400 MHz): ẟ (ppm) 13.84 (s, 1H), 6.78 (s, 1H), 6.40 (s, 1H), 6.26 (s, 1H), 5.98 (s, 1H), 3.83 (s, 3H), 3.36–3.24 (m, 2H), 2.69–2.59 (m, 2H), 1.79–1.54 (m, 2H), 1.49–1.37 (m, 4H), 1.03–0.91 (m, 12H). APT (100 MHz, CDCl_3_): ẟ (ppm) 182.0 (C-9), 161.1 (C-1), 159.9 (C-3), 155.8 (C-6), 154.3 (C-4′),154.3 (C-5′), 142.4 (C-7), 139.3 (C-8), 112.2 (C-8′), 110.9 (C-2), 103.7 (C-9′), 101.2 (C-5), 92.6 (C-4), 62.3 (C-15), 40.2 (C-11′), 37.9 (C-11), 28.8 (C-12), 28.2 (C-12′), 25.5 (C-10′), 22.5 (C-13 y C-14), 22.5 (C-13′ y C-14′), 20.1 (C-10). The spectroscopic data were consistent with those reported in the literature for tetrahydro-α-mangostin [50].

3,6-di-pentoxy-α-mangostin (**10**). **4** was subjected to an O-alkylation reaction in the presence of 1-bromopentane, K_2_CO_3_ and DMF [65]. To a mixture of **4** (80 mg, 0.200 mmol) and 1-bromopentane (0.410 mL, 3.29 mmol), was added K_2_CO_3_ (691 mg, 5.05 mmol) in 3.0 mL of anhydrous DMF. The mixture of reaction was stirred for 24 h at room temperature and after this time the solvent was removed by distillation under reduced pressure. Subsequently, cold water is added and an extraction with CHCl_3_ (3 × 15 mL) was performed. The combined organic phases were filtered and dried with anhydrous Na_2_SO_4_. The crude product resulting from the vacuum concentration was purified by FC eluted with hexane:AcOEt (90:10), to give **10** (72 mg, 67.2% yield). Compound **10**: yellow solid, m.p.: 176–178 °C. ^1^H-NMR (CDCl_3_, 400 MHz): ẟ (ppm) 13.53 (s, 1H), 6.70 (s, 1H), 6.28 (s, 1H), 5.28 (q, *J* = 4.4 Hz, 2H), 4.15 (d, *J* = 6.5 Hz, 2H), 4.06 (dt, *J* = 17.2, 6.5 Hz, 4H), 3.83 (s, 3H), 3.38 (d, *J* = 7.1 Hz, 2H), 1.93–1.84 (m, 6H), 1.83 (s, 3H), 1.71 (s, 6H), 1.48 (ddd, *J* = 29.3, 15.0, 7.5 Hz, 10H), 0.98 (t, *J* = 7.2 Hz, 6H). APT (100 MHz, CDCl_3_): ẟ (ppm) 181.9 (C-9), 162.8 (C-1), 159.7 (C-3), 157.4 (C-6), 155.2 (C-4′y 5), 144.0 (C-7), 137,5 (C-8), 131.5 (C-12), 131.2 (C-12′), 123.4 (C-11), 122.5 (C-11′), 111.8 (C-8′), 111.4 (C-2), 103.8 (C-2), 98.6 (C-5′), 89.2 (C-4), 68.8 (C-16), 68.4 (C-16′), 60.7 (C-15), 28.6 (C-18), 28.6 (C-18¨), 28.2 (C-19), 28.2 (C-14¨), 28.2 (C-13′), 26.2 (C-17), 26.1 (C17¨), 25.9 (C-14), 25.8 (C-13), 18.1 (C-14), 17.8 (C-13), 14,0 (C-20, C-20′). The spectroscopic data were consistent with those reported in the literature for 3,6-di-pentoxy-α-mangostin [51].

Fuscaxanthone C (**6**) and 3,6-di-methoxy-4-methyl-α-mangostin (**11**). The methylation reaction on **4** was also carried out in the presence of methyl iodide and oxide silver [67,68]. CH_3_I (1.5 mL, 13.0 mmol) was added to a mixture of **4** (50 mg, 0.125 mmol), AgO (250 mg, 1.07 mmol) and anhydrous acetone (5 mL). The mixture was stirred at room temperature for 2h and then excess CH_3_I was removed by distillation under pressure. Then, a liquid-liquid extraction was performed with AcOEt (4 × 5 mL) and the organic phases combined were washed with saturated NaCl solution (3 × 5 mL), and dried with anhydrous Na_2_SO_4_. The crude product resulting from the vacuum concentration was purified by FC with hexane:AcOEt (90:10), to give **11** (8.7 mg, 15.3% yield) and **6** (89.3 mg, 39.4% yield). Compound **11**: yellow solid, m.p.: 90–92 °C. ^1^H-NMR (CDCl_3_, 400 MHz): ẟ (ppm) 13.46 (s, 1H), 6.80 (s, 1H), 5.25–5.20 (m, 2H), 4.13 (d, J = 6.7 Hz, 3H), 3.99 (s, 3H), 3.80 (d, *J* = 2.9 Hz, 6H), 3.40 (d, *J* = 6.9 Hz, 2H), 2.32 (s, 3H), 1.85 (d, *J* = 1.3 Hz, 4H), 1.81 (d, *J* = 1.3 Hz, 6H). APT (100 MHz, CDCl_3_): ẟ (ppm) 182.8 (C-9), 162.8 (C-1), 158.6 (C-3), 158.3 (C-6), 155.6 (C-5′), 152.3 (C-4′), 144.0 (C-7), 137.3 (8), 131.8 (C-12′), 131.6 (C-12), 123.1 (C-11′), 122.8 (C-11), 116.6 (C-2), 111.8 (C-8′), 107.6 (C-4), 106.2 (C-9′), 98.3 (C-5), 61.0 (C-16), 60.9 (C-17), 56.0 (C-15), 26.2 (C-10), 25.9 (C-13), C-14), 22.4 (C-10′), 18.2 (C-14), 17.9 (C-13), 8.5 (C-18). The spectroscopic data were consistent with those reported in the literature for 3,6-di-methoxy-4-methyl-α-mangostin [52].

### 3.5. Determination of Enzymatic Inhibition against PL, AA and AG

#### 3.5.1. PL Inhibition Assay

Determination of enzymatic inhibition of the identified compounds against the PL enzyme was carried out following the methodology described in the literature with some modifications [39,69]. The concentrations evaluated for extract and fractions were from 1000 to 6.25 ppm and for compounds from 1000 to 6.25 µM. Orlistat was used as a positive control. TIn 96-well boxes, 30 μL of extract or compound stock solution, 30 μL of PL solution (200 U/mL) and 160 μL of Tris-HCl buffer (0.1 M, pH: 8.4) were mixed. The mixture obtained was pre-incubated for 30 min at 37 °C. Subsequently, 30 μL of p-nitrophenyl dodecanoate (100 μM) were added to complete a final volume of 250 μL and incubated for 40 min at 37 °C. After the incubation time, the absorbance was measured at 405 nm. All assays were performed in triplicate in two independent assays. IC_50_ values were estimated by nonlinear regression analysis. 

#### 3.5.2. AA Inhibition Assay

Determination of enzymatic inhibition of the identified compounds against the AA enzyme was performed following the methodology described in the literature with some modifications [39,70]. The concentrations evaluated for extract and fractions were from 1000 to 6.25 ppm and for compounds from 1000 to 6.25 µM. Acarbose was used as a positive control. In 96-well boxes, 50 μL of extract or compound stock solution and 50 μL of AA solution (2 U/mL) were mixed. The mixture obtained was pre-incubated for 30 min at 37 °C, and after this time, 50 μL of starch (5%) and incubated for 60 min at 37 °C. Finally, 100 μL of 3,5-dinitrosalicilic acid (DNS, 0.04 M) was added and heated for 10 min at 100 °C. After the reaction time, the absorbance at 540 nm was measured. All assays were performed in triplicate in two independent experiments. IC_50_ values were estimated by nonlinear regression analysis. 

#### 3.5.3. AG Inhibition Assay

Determination of enzyme inhibition of the compounds identified against the AG enzyme was carried out following the methodology described in the literature with some modifications [39]. The concentrations evaluated for extract and fractions were from 1000 to 6.25 ppm and for compounds from 1000 to 6.25 µM. Acarbose was used as a positive control. In 96-well boxes, 10 μL of stock solution of each extract or compound, 20 μL of GA solution (0.5 U/mL) and 200 μL of phosphate buffer (0.1 M, pH: 7.2) were mixed. The mixture obtained was pre-incubated for 30 min at 37 °C. Subsequently, 20 μL p-nitrophenyl-α-D-glucopyranoside (400 μM) was added to complete a final volume of 250 μL and incubated for 30 min at 37 °C. After the reaction time, the absorbance at 405 nm was measured. All assays were performed in triplicate in two independent experiments. IC_50_ values were estimated by nonlinear regression analysis. 

### 3.6. Kinetic Study

Kinetic studies to determine the type of inhibition were performed with PL, AA and AG using a methodology such as that described in the inhibitory activity assays. The xanthones were evaluated at 3 different concentrations according to their IC_50_ using the following values: 0.5, 1.0 and 2.0 × IC_50_. Five substrate concentrations were used for each of the enzymes in the range between 12.5 and 200 μM p-nitrophenyl dodecanoate for PL, 1 to 10% starch for AA and 25 to 800 μM for p-nitrophenyl-α-D-glucopyranoside in AG. Kinetic experiments were performed in triplicate in two independent experiments. Inhibition constants (Ki) were calculated from [substrate] vs reaction rate curves using nonlinear regression of the enzyme inhibition kinetic function. Additionally, the inhibition mechanism was graphically determined by applying the double reciprocal Lineweaver–Burk regression function [71,72].

### 3.7. Statistical Studies

Statistically significant differences in the biological effects of inhibitors (extracts, fractions, and compounds) were analyzed and compared using ANOVA, supplemented by Tukey HSD post hoc analysis. All reported data correspond to the average of three repetitions ± SD, the statistical significance considered was *p* < 0.05.

### 3.8. Molecular Docking Studies

Interaction, docking and binding analyses in 3D were performed using AutoDock4 (AD4), AutoDock Vina 1.1.2 (ADV) and Glide (Mae) software (Master Release-2016 from the Schrödinger platform) [73,74]. The crystal structures of PL, AA and AG were obtained from the Protein Data Bank website (www.rcsb.org, accessed on 22 February 2022), PDB ID: 1LPB, 4GQR and 2QMJ, respectively for each enzyme). The chemical structures (2D) of the ligands were processed by ChemDraw Professional. 16.0, obtaining the SMILES (Simplified Molecular Input Line Entry System) structures in its own database. Finally, and based on the experimental data of the type of inhibition, the most probable binding sites of the ligands (**1** to **11**) were evaluated, and the binding mode was analyzed and compared with the results obtained by each enzyme target. The Maestro academic software was used to generate 2D and 3D figures of the binding modes [74].

## 4. Conclusions

The present study contributes to the determination of prenylated xanthones from *G. mangostana* with action on the digestive enzymes pancreatic lipase (PL), α-amylase (AA) and α-glucosidase (AG). The mechanism of enzymatic inhibition of xanthones isolated and synthesized on the three digestive enzymes of interest was determined, allowing to establish that **2**, **4**, **5**, **6** and **9** are polyfunctional inhibitors of PL, AA and AG. Additionally, this is the first report of inhibitory activity against the three enzymes of compounds **6** to **11** and is the first report against AA of **1** to **3**. The results of this research demonstrate the potential of xanthones from *G. mangostana* as polypharmacological compounds of interest for the development of treatments for obesity and type 2 diabetes.

## Data Availability

Not applicable.

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
