# Peer review of "Multitarget Action of Xanthones from Garcinia mangostana against α-Amylase, α-Glucosidase and Pancreatic Lipase"

_molecules, 2022, doi:10.3390/molecules27103283_

Round 1

Reviewer 1 Report

The manuscript entitles “Multitarget action of xanthones from Garcinia mangostana against α-amylase, α-glucosidase and pancreatic lipase” has been written good and Authors planned very good research work inside the manuscript but have some comments below:

Line no 37,  Use comma after 2019.

Line no 48. Rewrite this sentence “A study developed by NCD Risk Factor Collaboration (NCD-RisC) in 2016 documents that in 1974 approximately 105 million adults worldwide were obese, and that by 2014 the prevalence of obesity multiplied by six, reaching 640 million adults in this condition”.

Authors need to explain why this study have you been planned in introduction part? I couldnot understand how authors can forgot this important things that as you are planning to know any problem after then objective should be given.

What do you mean about “bioguiaded chemical study?

Authors need to put values as they got from result 2.1 inside the result section. As authors only wrote that this treatment found better but how much, they should be mention .

Line no 142,  What do you mean about this sentence? “ 4 was subjected to a typical demethylation reaction using microwave irradiation in the presence of pyridine hydrochloride, however the expected product was not obtained”. Why you have written if expected product is not obtained?? please rewrite it.

Please give high quality of Figure 3, it should be readable.

In methodology, Statistical part should be separate.

Authors need to make tables with statistically approved as data is significant or non significant. Describe the data according inside the manuscript also as authors explain they did Tukeys test but not mention in table, don’t know why? Also differentiate the data according to Tukeys test.

How did the authors find results are significantly proved?

In Tables, Author should write the Different letters separately indicate significant differences (p < 0.05).

References should be according to the journals guideline. And follow same format for all the references.

English language should be improved in throughout the manuscript.

Author Response

Respected reviewer,

In the following document, we allow ourselves to highlight in yellow each point suggested for improvement in its review.

We add the data corresponding to the statistical tests, these had not previously been put in the tables, which can now be seen.

We have reviewed the style of the English language used.

Likewise, we review the references again as you did not suggest, we hope this version is more in accordance with those evaluated by you.

We greatly appreciate having taken the time to read it and make suggestions that strengthen our work.

Reviewer 2 Report

Dear Authors

congrats on this solid study. Some minor comments:

  • the introduction is mostly written towards diabetes; but the main topic is on xanthones ... here some more background would be supportive.
  • table 2; different digits in same colum; I think 1 is enough and a second "0" can be omitted for sure.
  • Fig2 is low in quality, this is just an overview, a zoom into binding sites would be beneficial

Author Response

Respected reviewer,

In the following document, we allow ourselves to highlight in yellow each point suggested for improvement in its review.

Initially, we reviewed the style of the English language used.

We have tried to expand the introduction a little more with those aspects that would be more important for the title of the article, we also took the trouble to improve the quality of the images and diagrams presented in our article to satisfy and present a more solid and better work. presented, finally, we correct the tables seeking to have the same number of decimals, and these are better presented now.

We greatly appreciate having taken the time to read it and make suggestions that strengthen our work.
